# Pharmacodynamic Evaluation of a Single Dose versus a 24-Hour Course of Multiple Doses of Cefazolin for Surgical Prophylaxis

**DOI:** 10.3390/antibiotics10050602

**Published:** 2021-05-19

**Authors:** Aaron Heffernan, Jowana Alawie, Steven C Wallis, Saiyuri Naicker, Santosh Adiraju, Jason A. Roberts, Fekade Bruck Sime

**Affiliations:** 1School of Pharmacy, The University of Queensland, Woolloongabba, Brisbane, QLD 4103, Australia; aaron.heffernan@griffithuni.edu.au (A.H.); jalawie011@gmail.com (J.A.); 2School of Medicine, Griffith University, Southport, Gold Cost, QLD 4222, Australia; 3University of Queensland Centre for Clinical Research, Faculty of Medicine, University of Queensland, Herston, Brisbane, QLD 4029, Australia; s.wallis@uq.edu.au (S.C.W.); s.naicker@uq.edu.au (S.N.); s.adiraju@uq.edu.au (S.A.); j.roberts2@uq.edu.au (J.A.R.); 4Department of Intensive Care Medicine and Pharmacy Department, Royal Brisbane and Women’s Hospital, Herston, Brisbane, QLD 4029, Australia; 5Division of Anaesthesiology Critical Care Emergency and Pain Medicine, Nîmes University Hospital, University of Montpellier, 30900 Nîmes, France

**Keywords:** cefazolin, pharmacodynamics, surgical prophylaxis, staphylococcus aureus

## Abstract

The optimal perioperative duration for the administration of cefazolin and other prophylactic antibiotics remains unclear. This study aimed to describe the pharmacodynamics of cefazolin for a single 2 g dose versus a 24 h course of a 2 g single dose plus a 1 g eight-hourly regimen against methicillin-susceptible *Staphylococcus aureus*. Static concentration time–kill assay and a dynamic in vitro hollow-fibre infection model simulating humanised plasma and interstitial fluid exposures of cefazolin were used to characterise the pharmacodynamics of prophylactic cefazolin regimens against methicillin-sensitive *Staphylococcus aureus* clinical isolates. The initial inoculum was 1 × 10^5^ CFU/mL to mimic a high skin flora inoculum. The static time–kill study showed that increasing the cefazolin concentration above 1 mg/L (the MIC) did not increase the rate or the extent of bacterial killing. In the dynamic hollow-fibre model, both dosing regimens achieved similar bacterial killing (~3-log CFU/mL within 24 h). A single 2 g dose may be adequate when low bacterial burdens (~10^4^ CFU/mL) are anticipated in an immunocompetent patient with normal pharmacokinetics.

## 1. Introduction

Cefazolin is one of the most common antibiotics used for surgical prophylaxis with in vitro activity against methicillin-susceptible *Staphylococcus aureus* (MSSA), which causes about 34% of all surgical site infections (SSIs) [1]. The optimal perioperative duration for the administration of cefazolin and other prophylactic antibiotics remains unclear. A systematic review of randomised controlled trials has suggested that administration of a single dose just before the procedure is sufficient to prevent SSIs compared with up to 48 h of antibiotic administration [2]. Indeed, a single dose of antibiotic administered prior to skin incision is advocated for in global guidelines [3]. However, a prolonged antibiotic prophylaxis may be used for some patients who are at a high risk of SSIs, such as those requiring cardiac, vascular, and orthognathic surgery [3,4]. Although both a single dose and a prolonged course of prophylaxis may both be observed in current practice, their relative effect in terms of maximising bacterial killing and suppressing regrowth of resistant sub-populations has not been demonstrated for commonly used antibiotics such as cefazolin. The aim of this study was to describe the pharmacodynamics of simulated unbound plasma and interstitial fluid (ISF) cefazolin exposures with a single 2 g dose versus a 24 h course of a single 2 g dose followed by a 1 g dose administered every 8 h against MSSA. 

## 2. Materials and Methods

### 2.1. Bacterial Isolates

One clinical MSSA isolate (#CTAP54) was used for all experiments. The isolate was suspended in 20% glycerol containing cation-adjusted Mueller-Hinton II broth (Ca-MH) and stored in a −80 °C freezer. The isolate was cultured on Ca-MH agar plates and incubated at 37 °C for 24 h for the experiment inoculum preparation. 

### 2.2. Antibiotic Agents

The analytical reference standard of cefazolin (FCB031845 Fluorochem) and a clinical formulation (batch number AFT171582) were used for in vitro susceptibility testing and HFIM dosing, respectively.

### 2.3. In vitro Susceptibility Testing

The cefazolin MICs were determined by broth microdilution [5,6]. In brief, serial two-fold dilutions of cefazolin were prepared in a Ca-MH broth in round bottom 96-well plates. Bacteria were suspended in the Ca-MH broth to a final inoculum of 1 × 10^5.5^ CFU/mL. Inoculated plates were incubated for 16 to 20 h at 37 °C. The lowest concentration of cefazolin that completely inhibits growth as detected by the unaided eye was recorded as the MIC.

### 2.4. Static Time–Kill Model

Preliminary evaluation of the bacterial killing pattern of increasing concentrations of cefazolin ranging from 0.5 to 20-fold as the MIC was performed in a static concentration time–kill study over 48 h with an initial bacterial concentration of 10^6^ CFU/mL. Experiments were conducted at 37 °C. After 24 h, the bacterial suspension was centrifuged and resuspended in fresh Ca-MH broth containing the relevant cefazolin concentration. Samples were taken at 0, 2, 4, 6, 8, 10, 24, 26, 30, and 46 h for quantification on the Ca-MH agar and the cefazolin-supplemented Ca-MH agar at a concentration fourfold the baseline isolate MIC. The sample was washed twice in sterile phosphate-buffered saline and appropriately diluted prior to plating. The static time–kill study was performed in duplicate.

### 2.5. Hollow-Fibre Infection Model

The HFIM was assembled as previously described over 96 h [7,8]. One HFIM circuit was conducted for each dosing regimen (single 2 g dose or 2 g followed by 1 g eight hourly for 24 h, i.e., additional three doses) and site (plasma or ISF), with an initial inoculum of 1 × 10^5^ CFU/mL to mimic a high skin flora inoculum.

Cefazolin unbound plasma and ISF concentration–time profiles were simulated using previous pharmacokinetic studies in a patient with a creatinine clearance of 100 mL/min, weighing 80 kg with a simulated half-life, and the times that the maximum concentrations were observed were 2.6 and 0.08 h, respectively, for plasma, and 3.15 and 2.0 h, respectively, for the ISF [9]. It was assumed that cefazolin was administered 30 min prior to skin incision. Pharmacodynamic parameters for each dosing regimen are listed in Table 1. Samples for pharmacokinetic analysis were taken from the central compartment at predefined times and immediately stored in a −80 °C freezer until analysis. Bacterial samples were taken from the sampling port of the extra-capillary space of the HFIM cartridge throughout the experiment and washed with sterile phosphate-buffered saline before plating 100 μL of diluted suspension onto standard and antibiotic-impregnated Ca-MH agar at a concentration fourfold the baseline MIC. At the end of the experiment, the MIC of cefazolin was determined for bacteria surviving on day 4 after exposure to the test doses.

### 2.6. Drug Assay

Cefazolin was measured in Ca-MH broth using the liquid chromatography method. A 5 µL aliquot of the neat specimen was injected onto a Nexera2 UHPLC system with chromatographic separation achieved using an XBridge BEH C18 column (Milford, MA, USA) with a gradient of methanol 10% *v/v* in a phosphate buffer (10 mM at pH 6.3). Mean intra-batch accuracy and precision values were within 3.9% and 14.4%, respectively.

## 3. Results

The baseline MIC of cefazolin for the study clinical isolate #CTAP54, determined by broth microdilution method, was 1 mg/L. The static time–kill study showed that increasing the cefazolin concentration above 1 mg/L (the minimum inhibitory concentration, MIC) did not increase the rate or extent of bacterial killing (Figure 1).

The cefazolin concentration–time curve was appropriately simulated using the hollow-fibre infection model (HFIM; R^2^ 0.85; Figure 2 and Table 1).

Figure 3 shows the effect of unbound plasma and ISF exposures of cefazolin on the total bacterial burden of #CTAP54 over 5 days for the two simulated prophylactic regimens. Plasma exposures from both dosing regimens showed similar, rapid bacterial killing with a 3-log CFU/mL reduction of the initial bacterial load within 24 h. Compared to the 2 g single-dose regimen, the administration of three additional 1 g doses eight-hourly did not result in any further reduction of the total bacterial burden after 24 h. However, there was a relative delay in the regrowth following the 24 h course dosing compared with the 2 g single dose. On the other hand, ISF exposures rapidly reduced the bacterial burden to below the limit of quantification within 24 h with comparable rate and extent of killing for both dosing regimens. Rapid regrowth was noted after 24 h for the 2 g single dose ISF exposure, such that the total bacterial burden was increased to the growth control within 48 h. However, the rate of regrowth was delayed following the ISF exposure from the 2 g dose plus 1 g eight-hourly prophylactic regimen such that the total bacterial concentration increased to control bacterial density by 72 h. There was no appreciable difference in the rate or extent of killing between the plasma and ISF exposures between the single 2 g cefazolin dose and the single 2 g cefazolin dose followed by 1 g administered eight-hourly. However, the prolonged 24 h course delayed the onset of bacterial regrowth by ~24 h (Figure 3).

No emergence of resistance to cefazolin was observed after both a single dose and a 24 h course of multiple doses. All bacterial regrowth occurred ~24 h after cessation of the cefazolin administration and remained susceptible to cefazolin with an MIC of 0.5 mg/L in both plasma and ISF following a single dose and prolonged duration therapy. There was no visible growth on the drug-infused plates (4 mg/L) at all time points for all dosing regimens and exposures simulated.

## 4. Discussion

Our study evaluated the in vitro pharmacodynamics of cefazolin against a single MSSA isolate in the context of surgical prophylaxis. We have shown that the initial bacterial killing by exposures achieved in the plasma and ISF is approximately equivalent for a single dose versus a 24 h course of cefazolin. Prolonging the duration of antibiotic administration delays bacterial regrowth by approximately 24 h. No resistance emerged during our 5-day experiment.

Our static time–kill study findings are consistent with the time-dependent bactericidal action of beta-lactam antibiotics with minimal increases in bacterial killing at a concentration exceeding the MIC (Figure 1) [10]. Similar bacterial killing results were observed in a study by Jain et al. [11] who assessed the efficacy of humanised cefazolin tissue concentrations against MSSA in an in vitro pharmacodynamic model where cefazolin achieved a slow progression of bacterial burden reduction from 0 h to 22 h with a rapid regrowth after 22 h following a single dose of cefazolin. This would suggest that a single dose may be sufficient to prevent bacterial growth beyond the proposed time of most operations, allowing the immune response to reduce the probability of an SSI. Another key finding from the HFIM was that there was a lack of resistance emergence. The development of resistance to antibiotics by *S. aureus* mainly involves exchanging antimicrobial resistance genes via horizontal transfer between organisms [12]. Therefore, the likelihood of a de novo emergence of resistance with *S. aureus* in this isolated in vitro study is low.

A limitation to the in vitro HFIM is the absence of immune-mediated bacterial killing. However, the lack of immune function allows the direct estimation of the extent of antimicrobial activity that can be attributed to a drug and describes a worst-case scenario by providing a safety margin when the results are extrapolated to humans. This is important because *S. aureus* is able to survive within granulocytes [13]. However, according to a study by Drusano et al. [14], an initial bacterial burden of 1 × 10^5^ CFU/mL in granulocyte replete mice is unlikely to subsequently cause infection. Bacterial eradication of an infection source likely requires the antibiotics to reduce the bacterial burden to at least ~10^2^ CFU/mL within 24 h. Taken together, the currently proposed dosing target of cefazolin for surgical prophylaxis, whereby the cefazolin concentration should be maintained above the MIC throughout the procedure, is likely sufficient for most patients. A single 2 g dose achieves this target and is able to restrict bacterial growth where the initial inoculum is 1 × 10^5^ CFU/mL or below. A further limitation of this study is that the effects of different skin incision times were relative to the antibiotic administration. In this study, we simulated the antibiotic concentrations assuming the dose was administered 30 min before skin incision, in accordance with current guidelines. Therefore, our results may not generalise to other patients with different antibiotic administration times relative to the skin incision. For example, additional bacterial growth may occur if the antibiotic administration was delayed relative to the skin incision and may be an indication that additional antibiotic administration may be necessary.

An additional limitation of this study is that the simulated cefazolin exposure may not reflect all patient populations, limiting the external generalisability of our results in some patient subpopulations, such as obese patients. For example, a study by Brill et al. [15] evaluated a 2 g dose of cefazolin administered intravenously to obese and non-obese patients undergoing laparoscopic Toupet fundoplication surgery. The authors observed decreased tissue concentrations following the administration of a single dose of 2 g of cefazolin. This may compromise the overall bacterial killing and prophylactic outcome. Importantly, the observed cefazolin concentrations in this patient population are lower (<4 mg/L 240 min after a single 2 g dose) than that simulated in our study, where the cefazolin concentration in plasma and ISF remain >4 mg/L for approximately 10 h post-dose. A further limitation of our study is the simulation of only one creatinine clearance (100 mL/min). Patients with impaired renal function would likely have increased cefazolin, concentrations potentially improving bacterial killing. Conversely, patients with augmented renal clearance (creatinine clearance >130 mL/min) may rapidly clear cefazolin, leading to ISF concentrations below the pathogen MIC within four hours [16]. Thus, in such special patient population with potential altered pharmacokinetics, the antibacterial effect of exposures from different cefazolin prophylactic regimens should be evaluated.

In conclusion, a single 2 g dose of cefazolin appears to be equivalent to the extended 24 h course of 2 g of cefazolin plus 1 g of cefazolin every 8 h for surgical prophylaxis against MSSA and may be adequate, particularly when low bacterial burdens (approx. 10^4^ CFU/mL) are anticipated.

## Figures and Tables

**Figure 1 antibiotics-10-00602-f001:**
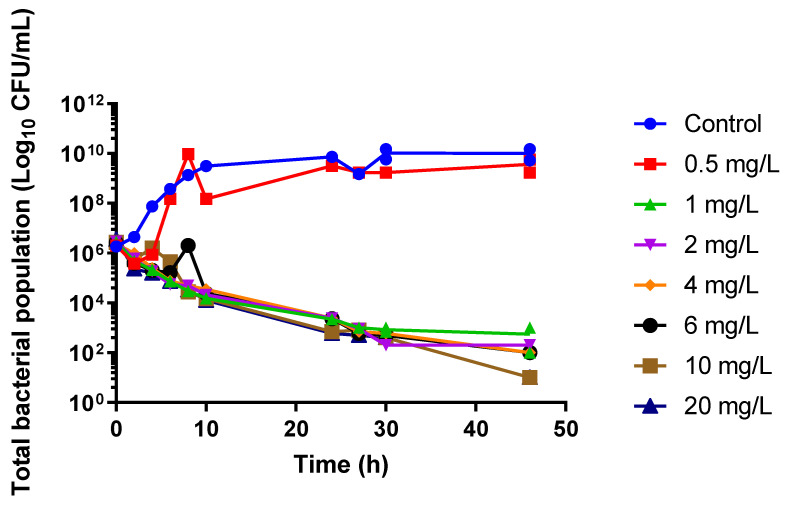
Static concentration time–kill curves for cefazolin against a methicillin-susceptible *Staphylococcus aureus* clinical isolate at seven different cefazolin concentrations. Connecting lines between data points represent the mean between the duplicate experiments.

**Figure 2 antibiotics-10-00602-f002:**
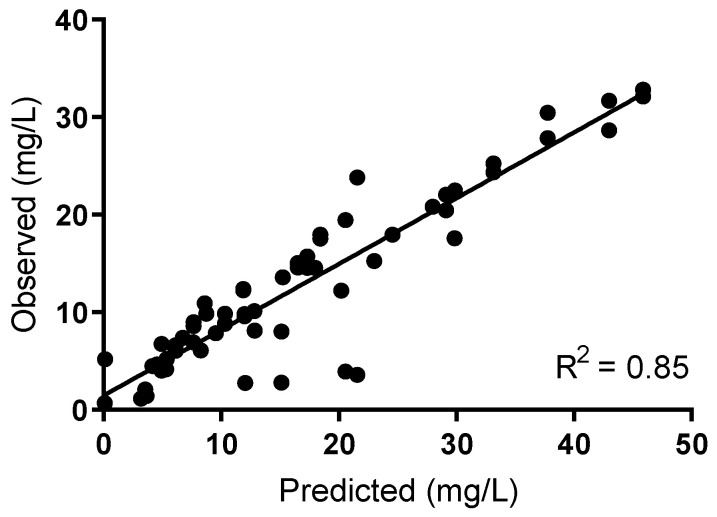
Cefazolin observed vs. predicted concentrations.

**Figure 3 antibiotics-10-00602-f003:**
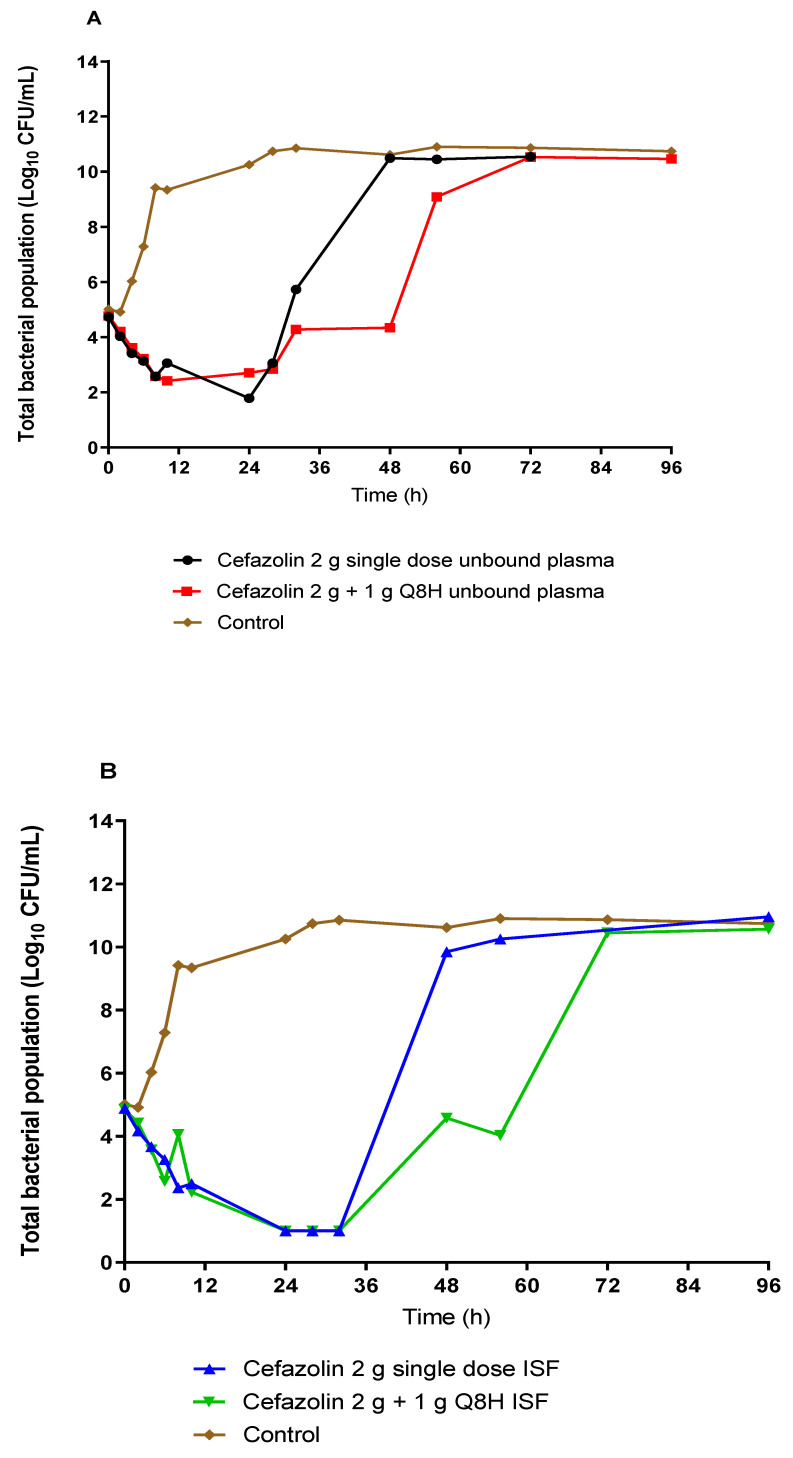
The effect of cefazolin prophylaxis on the total bacterial population of a methicillin-susceptible *Staphylococcus aureus* clinical isolate in a hollow-fibre infection model with simulated plasma (**A**) and ISF (**B**) unbound cefazolin exposures.

**Table 1 antibiotics-10-00602-t001:** Observed and expected pharmacodynamic parameters over the first 32 h of cefazolin for *S. aureus* #CTAP54.

Cefazolin Exposure	Expected			Observed		
	*f*AUC_0–32_	% *f*T > MIC _0–32 h_	*f*C_min_/MIC	*f*AUC_0–32_	% *f*T > MIC _0–32 h_	*f*C_min_/MIC
2 g single dose in plasma	164.75	46.09	0.01	145.21	38.57	<LOQ
2 g single dose in ISF	109.78	48.44	0.03	96.57	41.41	<LOQ
2 g plus 1 g 8-hourly in plasma	478.87	100	4.00	496.96	100	4.53
2 g plus 1 g 8-hourly in ISF	342.08	100	4.79	324.04	100	4.14

<LOQ, below level of quantification.

## Data Availability

Data is contained within the article.

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
