# Peer review of "Pharmacodynamic Evaluation of a Single Dose versus a 24-Hour Course of Multiple Doses of Cefazolin for Surgical Prophylaxis"

_antibiotics, 2021, doi:10.3390/antibiotics10050602_

Round 1
Reviewer 1 Report
Major Comments
- Lines 154-160: The use of 2g and 1g dosing as well as data from a study with 80 kg patients and CrCl 100 mL/min as the PK basis of the present study is limiting to the generalizability of the present study. While this is appropriately acknowledged in the discussion section, it warrants a more thorough discussion. Given the current state of obesity in the world, the scientific community should consider conducting studies based on the realistic human population. Many obese patients encountered may require 3 g doses of cefazolin based on prior research.
Minor Comments
Introduction: The introduction is thorough yet concise. Brief reference to worldwide guidelines on SSI prophylaxis would be welcomed as a note on how this research relates to current practice and recommendations, especially given the chosen dosing regimens in the present study.
Materials and Methods: Would like to see this section before the results and discussion. Overall, this section is well organized, easy to understand, and flows with the presentation of the results section.
- Lines 153-155: Please be more specific regarding dosing regimen, such as the total number of 1g doses administered over the 24 hour duration. Is the 24 hour time period the timing of last dose or the end of presumed coverage (i.e. 8 hours after the last dose, which may be >24 hours)?
- Line 161: “It was assumed that cefazolin was administered 30 minutes prior to skin incision.” Assumption should be noted as a limitation to the present study findings.
- Table 2: Would like to see re-created in the results section.
Results and Discussion:
- Recommend separating these sections in order to avoid introducing author biases into the results and confusion over results reporting. At the very least, recommend presenting all results prior to moving into discussion.
- Lines 87-95 and Table 1. This is a large portion of the results and discussion section, yet the comments on lines 93-95 would lead the reader to believe that the findings are not overwhelming relevant. Consider consolidating written findings and removing the associated table.
- Figure 3. Consider breaking into Figure 3: A (unbound plasma) and B (ISF) for better readability.
Author Response
Major Comments
Lines 154-160: The use of 2g and 1g dosing as well as data from a study with 80 kg patients and CrCl 100 mL/min as the PK basis of the present study is limiting to the generalizability of the present study. While this is appropriately acknowledged in the discussion section, it warrants a more thorough discussion. Given the current state of obesity in the world, the scientific community should consider conducting studies based on the realistic human population. Many obese patients encountered may require 3 g doses of cefazolin based on prior research.
Reply: We thank the reviewer for their comments on this manuscript. We agree that the issue of generalisability justifies further discussion. The discussion has been modified on lines 199-201 and 205-213 as highlighted below.
‘An additional limitation of this study is that the simulated cefazolin exposure may not reflect all patient populations limiting the external generalisability of our results in some patient subpopulations, such as obese patients. For example, a study by Brill et al. [16] evaluated a 2 g dose of cefazolin administered intravenously to obese and non-obese patients undergoing laparoscopic Toupet fundoplication surgery. The authors observed decreased tissue concentrations following the administration of a single dose of 2 g of cefazolin. This may compromise the overall bacterial killing and prophylactic outcome. Importantly, the observed cefazolin concentrations in this patient population are lower (<4 mg/L 240 minutes after a single 2 g dose) to that simulated in our study where the cefazolin concentration in plasma and ISF remain >4 mg/L for approximately 10 hours post-dose. A further limitation of our study is the simulation of only one creatinine clearance (100 mL/min). Patients with impaired renal function would likely have increased cefazolin, concentrations potentially improving bacterial killing. Conversely, patients with augmented renal clearance (creatinine clearance >130 mL/min) may rapidly clear cefazolin leading to ISF concentrations below the pathogen MIC within four hours [16].’
Minor Comments
Introduction: The introduction is thorough yet concise. Brief reference to worldwide guidelines on SSI prophylaxis would be welcomed as a note on how this research relates to current practice and recommendations, especially given the chosen dosing regimens in the present study.
Reply: We have included further discussion of the guidelines and added to the rationale for this study on lines 43-49 as highlighted below.
‘Indeed, a single dose of antibiotic administered prior to skin incision is advocated for in global guidelines [3]. However, a prolonged antibiotic prophylaxis maybe used for some patients who are at high risk of SSIs, such as those requiring cardiac, vascular and orthognathic surgery [3,4]. Although both single dose and prolonged course of prophylaxis may be observed in current practice, their relative effect in terms of maximising bacterial killing and suppressing regrowth of resistant sub-populations has not been demonstrated for commonly used antibiotics such as cefazolin .’
Materials and Methods: Would like to see this section before the results and discussion. Overall, this section is well organized, easy to understand, and flows with the presentation of the results section.
Reply: We have moved the results section to below the introduction as requested. However, we note that the journal article format places “materials and methods” section after the “discussion” section. If this is a strict requirement up on the journal’s editorial review, we are happy to amend as required.
Lines 153-155: Please be more specific regarding dosing regimen, such as the total number of 1g doses administered over the 24 hour duration. Is the 24 hour time period the timing of last dose or the end of presumed coverage (i.e. 8 hours after the last dose, which may be >24 hours)?
Reply: We have clarified this on line 87 – ‘i.e. additional three doses’.
Line 161: “It was assumed that cefazolin was administered 30 minutes prior to skin incision.” Assumption should be noted as a limitation to the present study findings.
Reply: We have added this as an additional limitation on lines 190-197 as highlighted below. ‘A further limitation of this study is that the effects of different skin incision times relative to the antibiotic administration. In this study, we simulated the antibiotic concentrations assuming the dose was administered 30 minutes before skin incision, in accordance with current guidelines. Therefore, our results may not generalise to other patients with different antibiotic administration times relative to the skin incision. For example, additional bacterial growth may occur if the antibiotic administration was delayed relative to skin incision and may be an indication where additional antibiotic administration may be necessary.’
Table 2: Would like to see re-created in the results section.
Reply: We have extensively revised this table by including the observed vs. predicted pharmacodynamic parameters and is included below for your reference.
Table 1: Observed and expected pharmacodynamic parameters over the first 32 h of cefazolin for S. aureus #CTAP54.
|
Cefazolin Exposure |
|
Expected |
|
Observed |
||
|
fAUC0-32 |
% fT>MIC 0-32 h |
fCmin/MIC |
fAUC0-32 |
% fT>MIC 0-32 h |
fCmin/MIC |
|
|
2 g single dose in plasma |
164.75 |
46.09 |
0.01 |
145.21 |
38.57 |
<LOQ |
|
2 g single dose in ISF |
109.78 |
48.44 |
0.03 |
96.57 |
41.41 |
<LOQ |
|
2 g plus 1 g 8-hourly in plasma |
478.87 |
100 |
4.00 |
496.96 |
100 |
4.53 |
|
2 g plus 1 g 8-hourly in ISF |
342.08 |
100 |
4.79 |
324.04 |
100 |
4.14 |
<LOQ, below level of quantification
Results and Discussion:
Recommend separating these sections in order to avoid introducing author biases into the results and confusion over results reporting. At the very least, recommend presenting all results prior to moving into discussion.
Reply: We have separated the results and discussion paragraphs. We have added additional comments on lines 158-167 to facilitate flow. ‘Our study has evaluated the in vitro pharmacodynamics of cefazolin against a single MSSA isolate in the context of surgical prophylaxis. We have shown that the initial bacterial killing by exposures achieved in the plasma and ISF is approximately equivalent for single dose versus 24 h course of cefazolin. Prolonging the duration of antibiotic administration delays bacterial regrowth by approximately 24 h. No resistance emerged during our 5-day experiment.
Our static time-kill study findings are consistent with the time-dependent bactericidal action beta-lactam antibiotics with minimal increases in bacterial killing at a concentration exceeding the MIC (Figure 1) [10].’
Lines 87-95 and Table 1. This is a large portion of the results and discussion section, yet the comments on lines 93-95 would lead the reader to believe that the findings are not overwhelming relevant. Consider consolidating written findings and removing the associated table.
Reply: We have removed the associated table and incorporated those results into the text as requested on line 162.
Figure 3. Consider breaking into Figure 3: A (unbound plasma) and B (ISF) for better readability.
Reply: We have amended as requested.
Reviewer 2 Report
- It is a very relevant topic, considering the need of increase the knowledge about optimal peri-operative duration of administration for cefazolin, especially for the surgical site infections high risk patients.
- It is a well written manuscript in a thematic of relevance and, despite the study limitations, the results can be useful, namely by the limited information about the optimal peri-operative duration of administration for cefazolin, especially for the surgical site infections high risk patients.
-
- The aim of the manuscript is to describe the pharmacodynamics of simulated unbound plasma and interstitial fluid cefazolin exposures from a single 2 g dose compared with a 24 h course of a single 2 g dose followed by 1 g administered 42 every 8 h, against methicillin-susceptible Staphylococcus aureus (MSSA).
- The authors found that a single 2 g dose of cefazolin appears to be equivalent to the extended 24-hour course of 2 g cefazolin plus 1 g cefazolin every 8 hours for surgical prophylaxis against MSSA.
- I consider the suitability of the presented manuscript for Antibiotics. It is a well written manuscript in a thematic of relevance and, despite the study limitations, the results can be useful, namely by the limited information about the optimal peri-operative duration of administration for cefazolin, especially for the surgical site infections high risk patients.
Author Response
We thank Reviewer 2 for their favourable review acknowledging the importance of the work.